# Evaluating Temporal Delays and Spatial Gaps in Overshoot-avoiding Mouse-pointing Operations

Shota Yamanaka*

Yahoo Japan Corporation

## ABSTRACT

For hover-based UIs (e.g., pop-up windows) and scrollable UIs, we investigated mouse-pointing performance for users trying to avoid overshooting a target while aiming for it. Three experiments were conducted with a 1D pointing task in which overshooting was accepted (a) within a temporal delay, (b) via a spatial gap between the target and an unintended item, and (c) with both a delay and a gap. We found that, in general, movement times tended to increase with a shorter delay and a smaller gap if these parameters were independently tested. Therefore, Fitts' law cannot accurately predict the movement times when various values of delay and/or gap are used. We found that 800 ms is required to remove negative effects of distractor for densely arranged targets, but we found no optimal gap.

**Index Terms:** H.5.2 [User Interfaces]: User Interfaces—Graphical user interfaces (GUI); H.5.m [Information Interfaces and Presentation]: Miscellaneous

## 1 INTRODUCTION

### 1.1 Background

In conventional studies on user performance in mouse-pointing tasks, users are instructed to point at and click an intended target as quickly and accurately as possible. The success or failure is judged by whether the mouse cursor falls within the target boundary when the mouse button is pressed, and the measured subjects (task results) are typically the movement time $MT$ and the error rate. Other measures of movement efficiencies have also been proposed by MacKenzie et al., such as Movement Variability [27]. In this paper, we revisit one of their proposed measures—Target Re-entry—for the case in which a cursor leaves the target and then enters it again. In particular, we focus on the case of overshooting, which is when the cursor passes through the target and then returns to it[1].

Such overshooting becomes problematic when, for example, a user is trying to select a target that pops up when the cursor hovers on a certain item, as shown in Fig. 1a. For example, in Amazon Prime Video, when users move the cursor onto a movie thumbnail, a corresponding window pops up to show the details of *Movie 3*: the description, users' rating, a *Play* button, etc. If users want to click on the *Play* button to start *Movie 3*, they should not largely overshoot it. If they do, the cursor leaves the pop-up window and hovers over the thumbnail of *Movie 1*, and unintentionally the pop-up window for *Movie 1* opens. Then, the user must try again by pointing to the thumbnail of *Movie 3* and aiming for the *Play* button.

Such an overshooting error can occur on the OS level. For example, as shown in Fig. 1b, when the cursor hovers over an application

*e-mail: syamanak@yahoo-corp.jp

---

[1]In some related papers, the term *overshoot* means that a click position falls beyond the target (e.g., [21]). In our paper, however, this term means that the cursor has crossed the target and has to move backward to return to

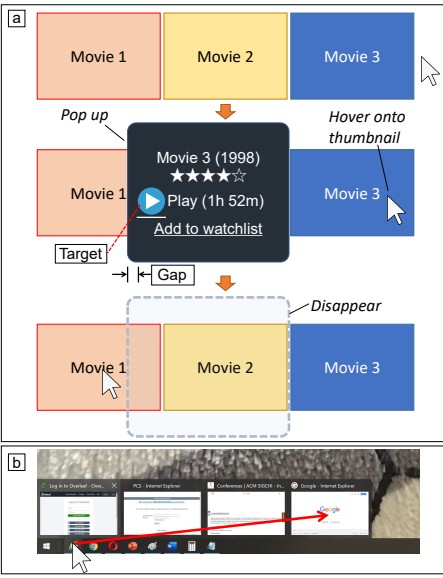

Figure 1: Examples of 2D pointing tasks where users want to avoid overshooting the target. (a) When the cursor hovers over the thumbnail of *Movie 3*, a window including a *Play* button pops up near the thumbnail, and a user wants to click on the button. The 'Gap' is the distance from the edge of the *Play* button to the edge of the pop-up window. (b) In Windows 10, multiple taskbar icons of a single application can be overlapped to save space. When the cursor hovers over an icon in the taskbar, all of the corresponding miniature windows are displayed, and users point to the intended one.

icon in the taskbar, all of the windows of that application pop up in miniature sizes, and the user points to an intended one. However, if the user overshoots it and a given time has passed, the miniature windows close. In other words, overshooting the target is permitted for a given duration.

As a more general case of overshooting error other than hover-based UIs, we introduce dragging to an intended position while avoiding page-scrolling. Fig. 2 shows examples of 1D tasks requiring dragging-and-dropping operations[2]. In Fig. 2a, a user wants to select multiple columns in a spreadsheet. If the user overshoots the aimed area and the cursor escapes from the window, the spreadsheet begins scrolling rapidly. The recovery time to go back to the initial view position is sometimes very long. Such unintended page-scrolling may occur on many applications including text editors as shown in Fig. 2b, drawing software, web browsers, file explorers, etc.

As shown in these examples, overshooting the target *during aiming* can be interpreted as an error operation in realistic UIs, although conventional pointing studies have considered only clicking outside the target as an error. The degree of care required to avoid overshooting depends on the task conditions. For example, when the *Gap* (or

---

[2]Dragging-and-dropping is also modeled by Fitts' law [15, 20, 28].

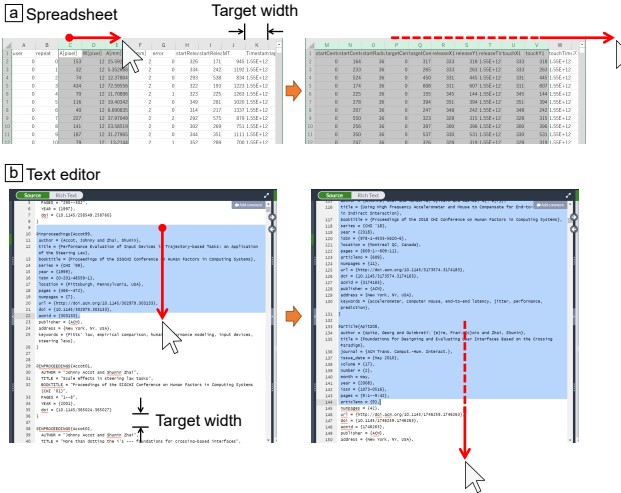

Figure 2: Examples of overshoot-avoiding 1D pointing tasks. (a) A user wants to select multiple columns from 'C' to 'K' in a spreadsheet. If the cursor overshoots the 'K' column and goes outside the window, the spreadsheet rapidly scrolls rightwards, and the user accidentally select the columns from 'C' to 'V'. Then, the user must go back to the initial view position by scrolling the spreadsheet leftwards. Note that vertical movements of the cursor are ignored when selecting multiple columns. (b) A user wants to select two blocks (from Line 9 to Line 37) in a BibTeX file on a text editor. If the cursor goes outside the window, the text editor begins scrolling and the user accidentally select more blocks from Line 9 to Line 143.

*offset*, *margin*) in Fig. 1a is wide, the possibility of unintentionally closing the pop-up window of *Movie 3* is small even if the user overshoots the target *Play* button. In the authors' environment, the *Gap* is 36 pixels, but how does the user performance change if the *Gap* is reduced to 18 pixels?

Similarly, as implemented on many websites and OSs, pop-up windows that close by cursor-leaving can have a temporal delay (or *timeout*). Such a delay, typically configured during website development by means of the `setTimeout` (JavaScript) or `delay` (jQuery) functions, gives users a chance to notice if accidental overshooting has occurred and then return the cursor to the pop-up window. This is desirable because it frees users from having to concentrate too hard on precise mouse operations. For example, the delay to close the miniature windows in Fig. 1b in our environment is approximately 400 ms, but we would have to point to the target window more carefully if it were 50 ms.

### 1.2 Research Question and Contribution Statement

It remains unclear how such gaps and delays help users. Possible drawbacks are, obviously, large gaps that take up a lot of screen space, and longer mouse cursor movements to point to intended items. For example, if the *Gap* in Fig. 1a is very wide, when the user wants to view the details of *Movie 1*, the cursor movement distance to the thumbnail increases. Because GUI designers carefully manage the space, unnecessary empty space should be avoided if users are not helped by the gaps.

Moreover, if the delay to close a pop-up window is too long, the user has to wait a long time for the window to close. For example, in Fig. 1a, when users want to view the description of *Movie 2* that is behind the pop-up window of *Movie 3*, they have to remove the cursor from the thumbnail and wait for the delay to finish. Also, in Fig. 1b, if users decide to click on a desktop icon behind the miniature windows, they have to remove the cursor from the taskbar

icon and the miniature windows and then wait for the given delay[3].

In summary, short delays and small gaps have potentially negative effects on user performance (i.e., by enforcing careful operation). At the same time, unnecessarily long delays and wide gaps should also be avoided. Therefore, to clarify how such delays and gaps affect mouse-pointing performance, we conducted three controlled experiments. In the first and second experiments, we evaluated the effects of delays and gaps independently. The third experiment was to test the interaction between delay and gap. For example, the pop-up window in Amazon Prime Video immediately closes when the cursor leaves the window, but if designers configure a delay of 200 ms, users can overshoot the target and then hover over an unintended object (in Fig. 1a, the thumbnail of *Movie 1*) for a short time without closing the pop-up window of *Movie 3*. This provides the chance to notice the overshooting and return to the intended target. Thus, users may be more relaxed when pointing to the *Play* button.

Our key contributions include:

**(1)** Conducting three experiments to evaluate the effects of delays and gaps in mouse-pointing tasks in which users have to avoid overshooting the target. The results showed that delays and gaps had different effects: the movement time *MT* decreased with large gaps, while the effect of delays plateaued for 400 ms or longer.

**(2)** Evaluating Fitts' law in all three experiments. Because delays and gaps significantly affected the *MT*, different intercepts and slopes should be used to accurately predict the *MT* depending on these two factors.

## 2 RELATED WORK

The problematic operation discussed in this paper, namely, overshooting a target, becomes an issue when indirect pointing devices are used. In contrast, when direct input methods such as finger touching are used, typically the system cannot sense overshooting of a finger above the surface.

In Fig. 1a, users have to avoid largely overshooting the target during a leftward horizontal movement, and they also must not deviate from the pop-up window on the y-axis. Such movements are called *steering* [1, 12, 33], but in this study we simplify our experiments by limiting these to 1D horizontal movements.

### 2.1 Effects of Timeout in GUI Operations

Regarding conventional desktop environments with mouse operations, timeouts have often been mentioned in relation to drop-down or cascaded menus for opening sub-menus by cursor-hovering over a menu item. That is to say, increasing the timeout can prevent unintended exposure of sub-menus, but a long timeout can force the user to wait to open intended sub-menus after hovering over the parent menu item [10, 23, 37, 38]. While these studies focused on improving menu operations by, for example, reducing the movement time and the error rate, the effects of timeout on the menu *pointing* time have not been modeled. Hence, we cannot refer to or modify the models for menu selection times. Bailly et al.'s recent survey [2] provides a thorough review of visual menu selection techniques. However, these studies do not directly answer our questions about how a timeout helps users' pointing operations with avoiding overshooting, even though such operations are required in typical PC tasks (Fig. 1 and Fig. 2).

Yamanaka proposed a refined version of steering law [1] with a timeout term to accept errors (deviation from a path) [47]. A difference in his work is that the necessary precision in path-steering tasks is higher than pointing; the cursor must not deviate from a path, and thus a timeout helps throughout the path. As a result, the *MT*

---

[3]In Windows 10, manually clicking on the desktop does not close the miniature windows.

was reduced by 54% at most in steering [47] but only by 17% at most in our experiments. Thus, designers should accept that the *MT* in pointing cannot be largely reduced even if a long timeout is set. As a series of work on modeling the *MT* with timeout, our work would be beneficial to model a targeted-steering task [11, 25, 35, 39], e.g., clicking a target after steering a path, such as menu selection. For example, in Fig. 1, users must not deviate from the pop-up window on the y-axis and then click the target. Our work opens up new research on such common tasks.

## 2.2 Effects of Gaps in Target Arrangements

If GUI items are densely arranged (i.e., small *Gap*), users tend to carefully point to the intended target. The effects of other unwanted items, hereafter *distractors*, have been studied. The area cursor [21] and its variations [9, 16] are typical examples affected by distractors, as the cursor size cannot be expanded among a dense group of distractors. A target-aware cursor jumping technique [17], target-aware gain-changing techniques [4, 44], and use of multiple cursors [24] are other examples affected by the distractor density. In the experiments undertaken for these studies, overshooting the target while aiming was permitted, as in the conventional Fitts' law task.

To model the density effect on these techniques, Blanch and Ortega introduced the *index of sparseness* of potential targets into Fitts' law [5]. Also in their model, overshooting while aiming was permitted, while in our experiments, such an operation is considered an error. Thus, the index of target sparseness could not be equivalently applied to our intended tasks in Fig. 1 and Fig. 2.

## 2.3 Performance Models of Pointing Tasks

For predicting user performance in aiming tasks, a promising model is Fitts' law [13] in the Shannon formulation [26, 36]:

$$MT = a + b \log_2 (A/W + 1) \quad (1)$$

where *MT* is the time to point to the target, *A* is the distance to the target, and *W* is its size. *a* and *b* are empirically determined constants. The logarithmic term is called the index of difficulty (*ID*):

$$ID = \log_2 (A/W + 1) \quad (2)$$

Fitts' law with nominal *A* and *W* values shows a good fit to error-free data [49].

Correcting the data when targets are missed has been discussed in previous works. In HCI, replacing *W* in Equation 2 with the effective width ($W_e$) is recommended [26, 36]. Here, $W_e$ is $4.133 \times \sigma$, where $\sigma$ is the standard deviation of the click positions.

As shown by Wright and Lee [45], however, the effective width method is not appropriate for our purpose. This is because, when designers set a target size to a new value (e.g., *W* = 30 pixels) after conducting a user study, they can estimate the *MT* for selecting the target by using *W* but cannot use $W_e$ without conducting a new user study. Thus, for predicting an average operation time when designing a GUI, using the nominal *ID* value is more appropriate, as was also mentioned by Zhai et al. [49]. More seriously, if many click positions fall outside a target, we obtain a larger $W_e$ value than the nominal *W*; e.g., $W_e$ = 34 pixels for a target of *W* = 30 pixels. This conflicts with our rule that the user must not overshoot a given target area when there is no gap. Thus, $W_e$ was not used in this study, and in our data analysis we use nominal values for Fitts' law.

## 3 EXPERIMENT 1: EFFECTS OF DELAY

We concurrently conducted Experiments 1 and 2 with the same participants on the same day. In Experiment 1, we evaluated the effects of a temporal delay in leaving then closing pop-up windows ($T_{delay}$), and in Experiment 2, we evaluated the effects of the *Gap*. Each experiment took 20 min to complete. The order of the two studies was counterbalanced among the 12 participants. This allowed

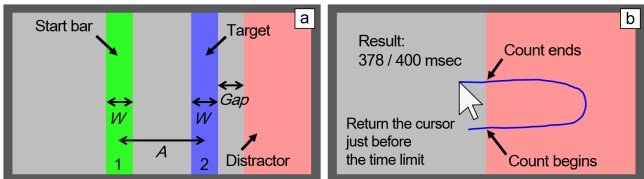

Figure 3: (a) Experimental parameter definitions. (b) Exercise system.

us to compare the results in Experiments 1 and 2. For example, if $T_{delay}$ significantly affected the *MT* but *Gap* did not, we can assume it is not due to any difference of the participant group. An experiment with 12 participants is common in the HCI field (particularly in the CHI literature [6]).

## 3.1 Participants

Twelve participants were recruited from a local university (all men; ages: *M* = 22.5, *SD* = 1.19 years). All had normal or corrected-to-normal vision, were right-handed, and were familiar with mouse operations. Three of them were daily mouse users. Each participant received 45 USD for his time (∼40 min total for Experiments 1 and 2).

## 3.2 Apparatus

The PC we used was a Sony Vaio Z (Core i7-5557U, 3.10 GHz, 4 cores; 16-GB RAM; Windows 10). The display was manufactured by Dell (2407WFPb: 24-inch diagonal, 1920 × 1200 pixel resolution, 518.4 × 324.0 mm display area, 3.70 pixels/mm; 16-ms response time; connected by an HDMI-to-DVI cable), and its refresh rate was set to 60 Hz. The input device was an iBuffalo optical mouse (BSMBU05: blue LED, 81.6 g, 1000 dpi; 1.5-m cable). We used a large mousepad (43 cm × 29 cm). The experimental system was implemented with Hot Soup Processor 3.4 and used in full-screen mode. The system read and processed input approximately 1000 times per second.

The mouse-cursor speed was set as the default in the OS, i.e., the control-display gain was in the middle of the slider. Pointer acceleration, denoted as the *Enhance pointer precision* setting in Windows 10, was enabled to allow the participants to perform mouse operations with higher ecological validity [8]. Using pointer acceleration does not violate a Fitts task (e.g., [46]), and it is consistent with the settings of a consumer OS such as Windows or macOS.

We adopted Müller et al.'s method [32] to measure the cursor latency from mouse movements. We used a Casio Exilim EX-ZR4000WE camera at 1000 fps. The experimenter moved the mouse rapidly by snapping his wrist (thus not mechanically controlled), and the mouse was hit with a heavy hard-cover book at high speed. The number of frames from when the mouse stopped to when the cursor stopped was counted. We repeated this action 30 times, and the average latency was 57.9 ms (*SD* = 11.2). This is in the range of typical mouse-display latencies of approximately 55 to 82 ms [7]. Therefore, we assume that the latency of our experimental system did not have a significant negative effect on user performance.

## 3.3 Task

We used discrete pointing tasks, as shown in Fig. 3a. First, participants clicked on the green start bar labeled "1", and then they clicked on the blue target bar labeled "2". If the cursor (a) overshot the target, (b) hovered over the pink distractor, and (c) had a hovering time above a given duration $T_{delay}$, then the trial was considered an error of closing the pop-up window $ER_{close}$. The $ER_{close}$ was differentiated from a typical error in pointing, or $ER_{click}$, which simply occurred when the click position was outside the target.

The participants were instructed to select the target as quickly as possible and to avoid making an error ($ER_{close}$ or $ER_{click}$). It did

not matter if the cursor returned to the target area within the given $T_{delay}$. In Experiment 1, there was no gap between the target and distractor ($Gap = 0$ pixel).

When an $ER_{close}$ occurred, the pink distractor turned red, and a "friction" sound was played. When an $ER_{click}$ occurred, a beep sound was played. Even if an erroneous operation was performed, the participants had to immediately aim for the target again; the task was not restarted from the beginning.

### 3.4  Design and Procedure

We tested six delay values ($T_{delay} = 0, 100, 200, 400, 800,$ and $\infty$ ms). Participants were not permitted any overshooting under the $T_{delay} = 0$ ms condition. The $T_{delay} = \infty$ condition, under which the participants did not have to worry about overshooting, was included to determine the baseline performance as in conventional Fitts tasks. The middle four values were selected on the basis of typical human reaction times. Because the time to correct hand movements according to visual feedback is longer than 200 ms [34]—approximately 260 [22] or 290 ms [31]—the participants could presumably return the cursor within ∼300 ms (including the system latency of 57.9 ms) after they noticed overshooting. Therefore, to observe the effects of delay, we set the $T_{delay}$ values to range from less than to sufficiently longer than human reaction time.

Two target distances ($A = 250$ and $600$ pixels) and three widths ($W = 25, 45,$ and $75$ pixels) were tested. When the $ID$ is smaller than approximately 3 or 4 bits, participants perform ballistic (feedforward) pointing motions [14, 19], such as aiming for a large or close target. Pointing to a close target is likely to occur in daily PC work, as shown in Fig. 1, and thus we set $ID$ to range from 2.7 to 5.6 bits to cover ballistic to visually controlled pointing motions. Note that the $ID$ here is the original formulation by Fitts: $ID = \log_2(2A/W)$ [13]. Because our main focus is on $T_{delay}$ and $Gap$, the numbers of $A$ and $W$ were relatively small.

One *block* consisted of a random order of $3A \times 2W \times 10$ repetitions $= 60$ trials with a fixed $T_{delay}$ value. The first repetition was considered practice. In addition, before each block, the participants used an exercise system (described below) to learn the delay of the next block, except for the $T_{delay} = 0$ ms and $\infty$ conditions. The order of the six $T_{delay}$ values was balanced among the 12 participants by using a Latin square. The movement direction was always to the right. In total, we recorded $3A \times 2W \times 9$ repetitions $\times 6T_{delay} \times 12$ participants $= 3888$ data points. After the all trials were completed, we interviewed the participants about the strategies in the experiments, such as how to shorten the $MT$ and reduce errors.

### 3.5  Exercise to Learn a Given Delay

As shown in Fig. 3b, the participants moved the cursor to the right pink area, and then a time measurement began. The goal was to return the cursor to the left gray area before the given $T_{delay}$. If the measured time was over the $T_{delay}$, the pink area turned red and a friction sound was played. The participants repeatedly performed this task and were expected to learn how to immediately return the cursor after entering the pink area. The session finished when a participant felt that he had sufficiently learned the $T_{delay}$ value. The time required was typically 30–40 sec (∼15 or 20 trials).

This exercise session allowed us to simulate a situation in which the participants were already familiar with the delay of a certain GUI. If we had not used this system, we would have been simulating participants using a GUI for the first time and learning the delay at every trial, but our purpose here was not to observe such a learning effect.

### 3.6  Results

We removed one spatial outlier data point (0.026%) whose movement distance was less than $A/2$ or whose click position was more than $2W$ from the target center [3]. For the remaining 3887

data points, we ran the normality tests (Shapiro-Wilk method with $\alpha = 0.05$) to the dependent variables. The results showed that the error-free $MT$ data did not normally distributed in 2 out of 36 conditions ($6T_{delay} \times 3A \times 2W$). Similarly, $ER_{click}$ rate, $ER_{close}$ rate, and $V_{peak}$ did not normally distributed in 36, 31, and 17 conditions, respectively. Hence, we used non-parametric ANOVAs with *Aligned Rank Transform* [43] with Tukey's *p*-value adjustment method for pairwise comparisons. Posthoc tests will be reported if we found main effects and first order interactions. The main effects of $T_{delay}$ on the $MT$, $ER_{click}$ rate, and $ER_{close}$ rate are shown in Fig. 4. After analyzing the speed profiles, we also decided to analyze the peak speed in a trial $V_{peak}$ as dependent variable to check how the participants changed their movement speeds depending on the task condition. The first-order interactions involving $T_{delay}$ are shown in Fig. 5.

#### 3.6.1  Movement Time $MT$

Again, when analyzing the $MT$ results and Fitts' law fitness, we used error-free data. We found significant main effects for $T_{delay}$ ($F_{5,55} = 11.34$, $p < 0.001$, $\eta_p^2 = 0.51$), $A$ ($F_{1,11} = 287.9$, $p < 0.001$, $\eta_p^2 = 0.96$), and $W$ ($F_{2,22} = 485.9$, $p < 0.001$, $\eta_p^2 = 0.98$). Posthoc tests showed significant differences (at least $p < 0.05$) for the following pairs on $T_{delay}$: (0, 100), (0, 200), (0, 400), (0, 800), (0, ∞), and (100, 800). Also for $W$, all the pairs showed significant differences with $p < 0.001$.

Significant interactions were found for $T_{delay} \times A$ ($F_{5,55} = 2.792$, $p < 0.05$, $\eta_p^2 = 0.20$) and $A \times W$ ($F_{2,22} = 14.01$, $p < 0.001$, $\eta_p^2 = 0.56$), but not for $T_{delay} \times W$ ($F_{10,110} = 1.365$, $p = 0.20$, $\eta_p^2 = 0.11$) and $T_{delay} \times A \times W$ ($F_{10,110} = 1.001$, $p = 0.45$, $\eta_p^2 = 0.08$). Posthoc tests showed significant differences (at least $p < 0.05$) for $A = 250$ pixels between the following $T_{delay}$ pairs: (0, 100), (0, 200), (0, 400), (0, 800), and (0, ∞). In the same manner, significant differences for $A = 600$ pixels between $T_{delay} = $ (0, 200), (0, 400), (0, 800), and (0, ∞). For all $W$ values, significant differences were found for the two $A$ values.

#### 3.6.2  $ER_{click}$ rate

Regarding conventional *pointing misses*, we observed 181 $ER_{click}$ trials (4.66%). We did not find significant main effects for $T_{delay}$ ($F_{5,55} = 0.8653$, $p = 0.51$, $\eta_p^2 = 0.073$), $A$ ($F_{1,11} = 0.3088$, $p = 0.59$, $\eta_p^2 = 0.027$), or $W$ ($F_{2,22} = 1.454$, $p = 0.26$, $\eta_p^2 = 0.12$). A significant interaction was found for $T_{delay} \times A \times W$ ($F_{10,110} = 2.041$, $p < 0.05$, $\eta_p^2 = 0.16$), but not for $T_{delay} \times A$, $T_{delay} \times W$, or $A \times W$ ($p > 0.05$ for all).

#### 3.6.3  $ER_{close}$ rate

We observed 167 $ER_{close}$ trials (4.30%) and found significant main effects for $T_{delay}$ ($F_{5,55} = 38.04$, $p < 0.001$, $\eta_p^2 = 0.76$), $A$ ($F_{1,11} = 39.45$, $p < 0.001$, $\eta_p^2 = 0.78$), and $W$ ($F_{2,22} = 38.87$, $p < 0.001$, $\eta_p^2 = 0.78$). Posthoc tests showed significant differences (at least $p < 0.05$) for the following pairs on $T_{delay}$: (0, 400), (0, 800), (0, ∞), (100, 400), (100, 800), (100, ∞), (200, 400), (200, 800), (200, ∞), (400, 800), and (400, ∞). Also for $W$, all the pairs showed significant differences (at least $p < 0.001$).

Significant interactions were found for $T_{delay} \times A$ ($F_{5,55} = 12.02$, $p < 0.001$, $\eta_p^2 = 0.52$), $T_{delay} \times W$ ($F_{10,110} = 10.52$, $p < 0.001$, $\eta_p^2 = 0.49$), $A \times W$ ($F_{2,22} = 23.97$, $p < 0.001$, $\eta_p^2 = 0.68$), and $T_{delay} \times A \times W$ ($F_{10,110} = 6.538$, $p < 0.001$, $\eta_p^2 = 0.37$). Posthoc tests showed significant differences (at least $p < 0.05$) for $A = 250$ pixels between the following $T_{delay}$ pairs: (0, 800), (0, ∞), (100, 800), (100, ∞), (200, 800), (200, ∞), (400, 800), and (400, ∞). In the same manner, significant differences for $A = 600$ pixels between $T_{delay} = $ (0, 400), (0, 800), (0, ∞), (100, 400), (100, 800), (100, ∞),

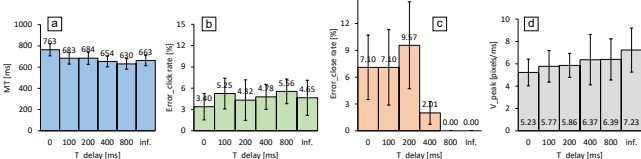

Figure 4: Main effects of $T_{delay}$ on the (a) $MT$, (b) $ER_{click}$ rate, and (c) $ER_{close}$ rate in Experiment 1. Error bars show 95% CIs.

(200, 400), (200, 800), and (200, ∞). For $W = 25$ and 75 pixels, the two $A$ pairs showed significant differences. For $W = 25$ pixels, significant differences on the following $T_{delay}$ pairs were found: (0, 800), (0, ∞), (200, 800), and (200, ∞). Similarly, for $W = 45$ pixels, (0, 800), (0, ∞), (100, 800), (100, ∞), (200, 800), and (200, ∞). Finally, for $W = 75$ pixels, (0, 800), (0, ∞), (100, 800), (100, ∞), (200, 400), (200, 800), (200, ∞), (400, 800), (400, ∞).

Note that $ER_{click}$ and $ER_{close}$ could occur concurrently in one trial (inclusive). The number of error-free trials was 3571, while 316 data points (8.13%) were removed because of errors found when analyzing the $MT$. This total error rate is greater than typical user experiments involving 1D target pointing (e.g., 4 or 5% [26, 42]), which is possibly because we defined new error criterion (i.e., $ER_{close}$).

### 3.6.4  Peak Speed $V_{peak}$

We found significant main effects for $T_{delay}$ ($F_{5,55} = 5.620$, $p < 0.001$, $\eta_p^2 = 0.34$), $A$ ($F_{1,11} = 328.5$, $p < 0.001$, $\eta_p^2 = 0.97$), and $W$ ($F_{2,22} = 4.500$, $p < 0.05$, $\eta_p^2 = 0.29$). Posthoc tests showed significant differences (at least $p < 0.05$) for the following pairs on $T_{delay}$: (0, ∞), and (100, ∞). A significant differences were only found between $W = 25$ and 45 pixels ($p < 0.05$).

Significant interactions were found for $T_{delay} \times A$ ($F_{5,55} = 4.256$, $p < 0.01$, $\eta_p^2 = 0.28$), $A \times W$ ($F_{2,22} = 5.374$, $p < 0.05$, $\eta_p^2 = 0.33$), and $T_{delay} \times A \times W$ ($F_{10,110} = 2.001$, $p < 0.05$, $\eta_p^2 = 0.15$), but not for $T_{delay} \times W$ ($F_{10,110} = 1.710$, $p = 0.087$, $\eta_p^2 = 0.13$). Posthoc tests showed significant differences (at least $p < 0.05$) for $A = 250$ pixels between the following $T_{delay}$ pairs: (0, 800), (0, ∞), and (100, ∞), but not for any $T_{delay}$ pairs for $A = 600$ pixels. For all $W$ values, significant differences were found for the two $A$ values. In summary, $T_{delay} = 0$ and 100 ms showed significantly slower peak speeds, yet $T_{delay} \geq 200$ ms did not contribute to increasing the peak speed any more.

### 3.6.5  Model Fitting

Fitts' law showed $R^2 > 0.99$ in each case for all $T_{delay}$ conditions by using $N = 6$ ($2A \times 3W$) data points as shown in Fig. 6. This means that the $MT$ can be accurately predicted if we use a single $T_{delay}$ value. However, if several $T_{delay}$ conditions are mixed, the prediction accuracy was comparatively low ($R^2 = 0.882$).

### 3.6.6  Speed Profiles

Next, we investigated how the $T_{delay}$ affected pointing behaviors during aiming. Fig. 7 shows the speed profiles for each $T_{delay}$ under the longer distance ($A = 600$ pixels) and the smallest target size ($W = 25$ pixels, which required the most careful cursor positioning to avoid overshooting). Because the raw data were very noisy, we re-sampled the cursor trajectories every 25 pixels. Because the clicked positions in the start and target bars varied in every trial, we show the speeds from when the cursor left the start bar to when it entered the target ($A - W = 575$ pixels in this case).

Although the speed profiles in Fig. 7 are still noisy, as the overall tendency, we can see two groups: speeds for longer than human reaction time ($T_{delay} \geq 400$ ms) were clearly higher than the other

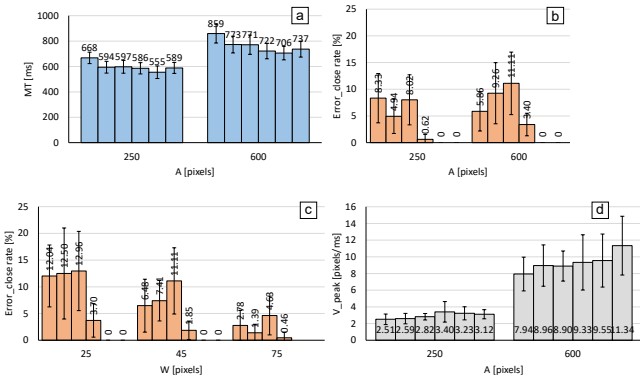

Figure 5: Interaction effects involving $T_{delay}$ on the (a) $MT$, (b–c) $ER_{close}$ rate, and $V_{peak}$ in Experiment 1. Error bars show 95% CIs.

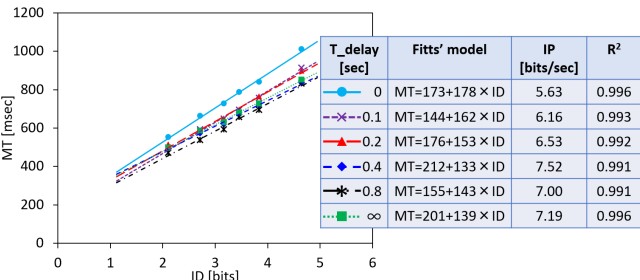

Figure 6: Fitts' law fitness in Experiment 1. The index of performance ($IP$) is $1/b$.

three conditions. This result indicates that a $T_{delay}$ helped the participants to accelerate the pointing speed, but if $T_{delay}$ was shorter than human reaction time, they could not effectively take advantage of such an overshoot-accepting condition.

Another important finding is that this benefit due to a long $T_{delay}$ increased the peak speeds, and a higher speed was maintained until entering the target area compared to the no-delay condition. Hence, we assume that the participants were more relaxed by a longer $T_{delay}$ not only during the final cursor positioning phase onto the target, but also throughout the feedback-loop controlling phase.

### 3.7  Discussion of Experiment 1

Regardless of the degree of care in avoiding overshooting due to the $T_{delay}$, Fitts' law showed $R^2 > 0.99$, while the $MT$ showed significant differences. Because we tested $T_{delay}$ ranging from 0 to ∞ ms, this high fitness will be observed for untested $T_{delay}$ values such as 1000 or 2000 ms. This result can help designers predict an $MT$ value for a given set of $A$ and $W$ if a $T_{delay}$ value is fixed. However, if designers want to use a new $T_{delay}$ value, they will have to conduct another user study, as Fitts' law for $N = 36$ data points showed $R^2 = 0.88$.

We had assumed that a longer $T_{delay}$ value would enable the participants to move the cursor more quickly. Yet, we found no significant difference in the $MT$ for $T_{delay}$ of 200 ms or longer. In addition, $T_{delay}$ showed no main effect on the $ER_{click}$ rate, and pair-wise tests of the $ER_{close}$ rate showed no significant differences for $T_{delay}$ of 800 ms or longer. Therefore, in summary, the upper $T_{delay}$ value to remove the negative effects of the overshoot-avoiding condition was 800 ms. For $T_{delay}$ of 800 ms or longer, no advantages for $MT$ or $ER_{close}$ rate would be gained. For $T_{delay}$ of 400 ms or shorter, negative effects to increase $ER_{close}$ rate would be observed.

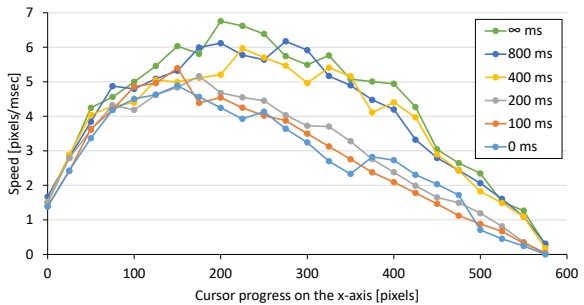

Figure 7: Speed profiles for each $T_{delay}$ where $A$ = 600 and $W$ = 25 pixels in Experiment 1.

## 4 EXPERIMENT 2: EFFECTS OF GAP

### 4.1 Design and Procedure

Experiment 2 was concurrently performed with Experiment 1 using the same participants and the same apparatus. We tested six *Gap* values (0, 8, 24, 72, 216, and ∞ pixels). This task simulated that the *Gap* area was still inside a pop-up window (as in Fig. 1a), and the pink distractor area was outside it. The participants were not permitted any overshooting under the *Gap* = 0 pixels condition. The conditions for *A*, *W*, and the movement direction were the same as in Experiment 1. Because $T_{delay}$ was fixed to 0 ms in Experiment 2, entering the pink distractor was not permitted, and the exercise system described for Experiment 1 was not used.

One *block* consisted of a random order of $3A \times 2W \times 6Gap$ = 36 trials. One block for practice and then nine blocks for data collection were performed. In total, we recorded $3A \times 2W \times 6Gap \times$ 9 repetitions × 12 participants = 3888 data points. After the all trials were completed, we interviewed the participants about the strategies in the experiments. The main effects of *Gap* are shown in Fig. 8. The first-order interactions involving *Gap* are shown in Fig. 9.

### 4.2 Results

#### 4.2.1 Movement Time (MT)

After removing two outlier data points (0.051%), we found significant main effects for *Gap* ($F_{5,55} = 13.34$, $p < 0.001$, $\eta_p^2 = 0.55$), $A$ ($F_{1,11} = 241.2$, $p < 0.001$, $\eta_p^2 = 0.96$), and $W$ ($F_{2,22} = 187.2$, $p < 0.001$, $\eta_p^2 = 0.94$). Posthoc tests showed significant differences for the following *Gap* pairs (at least $p < 0.05$): (0, 216), (0, ∞), (8, 72), (8, 216), (8, ∞), (24, 216), and (24, ∞). All *W* pairs showed significant differences with $p < 0.001$.

A significant interaction was found for $A \times W$ ($F_{2,22} = 10.83$, $p < 0.001$, $\eta_p^2 = 0.50$), but not for $Gap \times A$, $Gap \times W$, and $Gap \times A \times W$ ($p > 0.05$). For all *W* values, significant differences were found for the two *A* values.

#### 4.2.2 $ER_{click}$ rate

We observed 208 $ER_{click}$ trials (5.35%). We found significant main effects for *Gap* ($F_{5,55} = 2.955$, $p < 0.05$, $\eta_p^2 = 0.21$) and $W$ ($F_{2,22} = 14.00$, $p < 0.001$, $\eta_p^2 = 0.56$), but not for $A$ ($F_{1,11} = 2.504$, $p = 0.14$, $\eta_p^2 = 0.19$). Posthoc tests showed a significant difference between *Gap* = 0 and ∞ with $p < 0.05$. The *W* pairs of (25, 75) and (45, 75) showed significant differences with $p < 0.05$.

Significant interactions were found for $Gap \times A$ ($F_{5,55} = 2.430$, $p < 0.05$, $\eta_p^2 = 0.18$), $Gap \times W$ ($F_{10,110} = 4.243$, $p < 0.001$, $\eta_p^2 = 0.28$), and $Gap \times A \times W$ ($F_{10,110} = 3.405$, $p < 0.001$, $\eta_p^2 = 0.24$), but not for $A \times W$ ($F_{2,22} = 1.937$, $p = 0.17$, $\eta_p^2 = 0.15$). For the $Gap \times A$ and $Gap \times W$ interactions, posthoc tests showed no *Gap*

pairs having significant differences (it is possible for pairwise comparisons).

#### 4.2.3 $ER_{close}$ rate

We observed 131 $ER_{close}$ trials (3.37%). The number of error-free trials was 3569, while 317 data points (8.16%) were removed because of errors found when analyzing the *MT*.

We found significant main effects for *Gap* ($F_{5,55} = 32.86$, $p < 0.001$, $\eta_p^2 = 0.75$), $A$ ($F_{1,11} = 53.31$, $p < 0.001$, $\eta_p^2 = 0.83$), and $W$ ($F_{2,22} = 28.03$, $p < 0.001$, $\eta_p^2 = 0.72$). Posthoc tests showed significant differences for the following *Gap* pairs (at least $p < 0.05$): (0, 72), (0, 216), (0, ∞), (8, 72), (8, 216), (8, ∞), (24, 216), (24, ∞), (72, 216), and (72, ∞). The *W* pairs of (25, 75) and (25, 45) showed significant differences with $p < 0.05$.

Significant interactions were found for $Gap \times A$ ($F_{5,55} = 8.226$, $p < 0.001$, $\eta_p^2 = 0.43$), $Gap \times W$ ($F_{10,110} = 6.289$, $p < 0.001$, $\eta_p^2 = 0.36$), $A \times W$ ($F_{2,22} = 18.87$, $p < 0.001$, $\eta_p^2 = 0.63$), $Gap \times A \times W$ ($F_{10,110} = 4.163$, $p < 0.001$, $\eta_p^2 = 0.27$). Posthoc tests showed significant differences (at least $p < 0.05$) for $A$ = 250 pixels between the following *Gap* pairs: (0, 72), (0, 216), (0, ∞), (8, 72), (8, 216), (8, ∞), (24, 216), (24, ∞). Similarly, for $A$ = 600 pixels, the following *Gap* pairs showed significant differences: (0, 216), (0, ∞), (8, 216), (8, ∞), (24, 216), and (24, ∞). For *W* = 25 pixels, the following *Gap* pairs showed significant differences: (0, 216), (0, ∞), (8, 216), (8, ∞), (72, 216), (72, ∞). Similarly, for *W* = 45 pixels: (0, 8), (0, 72), (0, 216), (0, ∞), (24, 72), (24, 216), and (24, ∞). Finally, for *W* = 75 pixels: (0, 72), (0, 216), (0, ∞), (8, 72), (8, 216), (8, ∞), (24, 72), (24, 216), and (24, ∞). For all *W* values, significant differences were found for the two *A* values.

#### 4.2.4 Peak Speed $V_{peak}$

We found significant main effects for $T_{delay}$ ($F_{5,55} = 10.06$, $p < 0.001$, $\eta_p^2 = 0.48$), $A$ ($F_{1,11} = 206.2$, $p < 0.001$, $\eta_p^2 = 0.95$), and $W$ ($F_{2,22} = 7.952$, $p < 0.01$, $\eta_p^2 = 0.42$). Posthoc tests showed significant differences (at least $p < 0.05$) for the following pairs on *Gap*: (0, 8), (0, 216), (0, ∞), (8, 24), (24,216), (24, ∞), and (72, ∞). A significant differences were only found between *W* = 45 and 75 pixels ($p < 0.01$).

Significant interactions were found for $Gap \times A$ ($F_{5,55} = 6.452$, $p < 0.001$, $\eta_p^2 = 0.40$), $Gap \times W$ ($F_{10,110} = 2.384$, $p < 0.05$, $\eta_p^2 = 0.18$), $A \times W$ ($F_{2,22} = 5.165$, $p < 0.05$, $\eta_p^2 = 0.32$), and $Gap \times A \times W$ ($F_{10,110} = 2.432$, $p < 0.05$, $\eta_p^2 = 0.18$). Posthoc tests showed significant differences (at least $p < 0.05$) for $A$ = 250 pixels between the following *Gap* pairs: (0, 8), (0, ∞), (8, 24), (24, ∞), and (72, ∞), but not for any *Gap* pairs for $A$ = 600 pixels. Regarding the $Gap \times W$ interaction, we found two Gap pairs having significant differences: (0, ∞) and (24, ∞) for *W* = 45 pixels. For all *W* values, significant differences were found for the two *A* values ($p < 0.001$).

#### 4.2.5 Model Fitting

Fitts' law showed $R^2 > 0.98$ for all *Gap* conditions by using $N = 6$ data points, as shown in Fig. 10. When we did not separate the *Gap* conditions, the fit using $N = 36$ data points was $R^2 = 0.940$.

#### 4.2.6 Speed Profiles

Fig. 7 shows the speed profiles for each *Gap* under $A = 600$ and $W = 25$ pixels. Compared with the speed profiles in Experiment 1, we could not see clear differences between *Gap* values, while wider *Gap* values (216 and ∞ pixels) seemed to slightly increase the speed. This resulted in slight improvements in *MT* due to *Gap* values.

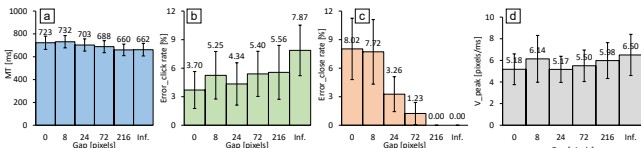

Figure 8: Main effects of *Gap* on the (a) *MT*, (b) $ER_{click}$ rate, and (c) $ER_{close}$ rate in Experiment 2.

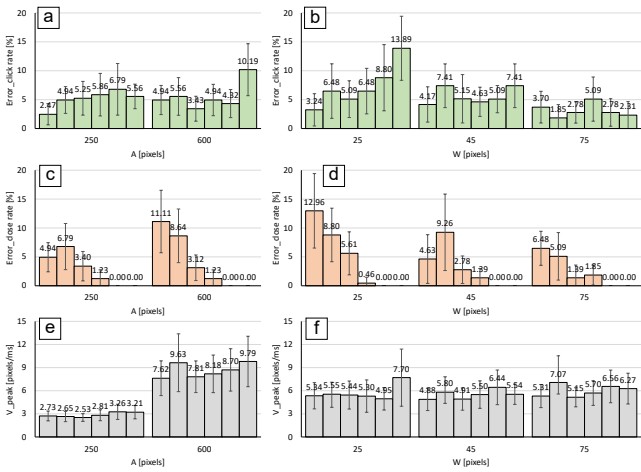

Figure 9: Interaction effects involving *Gap* on the (a–b) $ER_{click}$ rate, (c–d) $ER_{close}$ rate, and (e–f) $V_{peak}$ in Experiment 2. Error bars show 95% CIs.

### 4.3 Discussion of Experiment 2

Similarly to Experiment 1, as *Gap* increased, *MT* tended to decrease (Fig. 8a) and the $ER_{close}$ rate decreased (Fig. 8c). There were no significant differences for $Gap \geq 72$ pixels in *MT*, yet the $ER_{close}$ rate for $Gap = 72$ pixels was still significantly higher than that for $Gap = 216$ pixels. Thus, $Gap = 216$ pixels was needed to eliminate negative effects on user performance in our experimental conditions.

The model fitness without separating the *Gap* values was $R^2 = 0.94$, so designers can predict the average *MT* under a given condition regardless of the *Gap* with a certain degree of accuracy. Because we used the same participants in both Experiments 1 and 2 and the order of the two studies was counterbalanced, we conclude that the resultant differences probably stem from whether the "allowance" of overshooting was given temporally ($T_{delay}$) or spatially (*Gap*).

## 5 EXPERIMENT 3: INTERACTIONS OF DELAY AND GAP

In Experiment 2, we found that the *MT* decreased as the *Gap* increased (Fig. 8a). If the $T_{delay}$ and *Gap* were specified concurrently, however, the result might be different. For example, when the $T_{delay}$ is sufficiently long (e.g., 800 ms), there is little concern about invoking $ER_{close}$ (Fig. 4c), and hence, the positive effect of a large *Gap* should disappear. To confirm such potential interactions between the $T_{delay}$ and *Gap*, we conducted Experiment 3, which took 25 to 30 min per participant.

### 5.1 Apparatus and Participants

Experiment 3 was conducted two weeks after Experiments 1 and 2. The same apparatus was used. Twelve participants were again recruited from a local university (four women, eight men; ages: *M* = 22.2, *SD* = 1.72 years). All had normal or corrected-to-normal vision and were right-handed. Two of them were daily mouse users. Three of them had also participated in Experiments 1 and 2. Each participant received 23 USD in compensation.

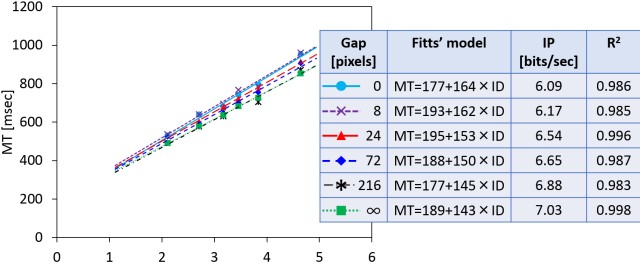

Figure 10: Fitts' law fitness in Experiment 2.

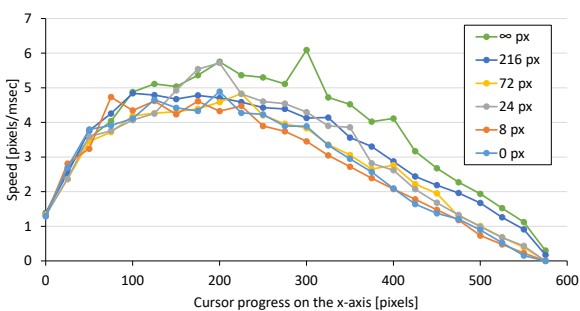

Figure 11: Speed profiles for each *Gap* where *A* = 600 and *W* = 25 pixels in Experiment 2.

### 5.2 Design and Procedure

Because the goal of Experiment 3 was to observe potential interactions between the $T_{delay}$ and *Gap*, we did not evaluate any baseline performance (i.e., the cases of $T_{delay} = \infty$ ms and $Gap = \infty$ pixels). If we had included $T_{delay} = \infty$, there would be no concern about overshooting regardless of the *Gap* values, and likewise for $Gap = \infty$. Also, because $T_{delay} = 800$ ms showed no significant difference from $T_{delay} = \infty$ for any results (*MT*, $ER_{click}$, and $ER_{close}$; Fig. 4), we used $T_{delay} = 400$ ms as the upper value. Thus, we reused four $T_{delay}$ values (0, 100, 200, and 400 ms) and five *Gap* values (0, 8, 24, 72, and 216 pixels). The *A* and *W* values were the same as in Experiment 1, and the movement direction was always to the right.

One *block* consisted of a random order of $3A \times 2W \times 5Gap \times 4$ repetitions = 120 trials with a fixed $T_{delay}$ value. Before each block, the participants used the exercise system as in Experiment 1, and then ten trials randomly selected from the 30 ($3A \times 2W \times 5Gap$) conditions were performed as practice. In total, we recorded $3A \times 2W \times 5Gap \times 4$ repetitions $\times 4T_{delay} \times 12$ participants = 5760 data points. After the all trials were completed, we interviewed the participants about the strategies in the experiments.

### 5.3 Results

We removed two outlier data points (0.035%). The main effects of $T_{delay}$ and *Gap* are shown in Fig. 12. The first-order interactions involving $T_{delay}$ and *Gap* are shown in Fig. 13. The number of error-free trials was 5389, while 369 data points (6.41%) were removed because of errors found when analyzing the *MT*.

#### 5.3.1 Movement Time (MT)

We found significant main effects for *Gap* ($F_{4,44} = 17.55$, $p < 0.001$, $\eta_p^2 = 0.61$), *A* ($F_{1,11} = 708.7$, $p < 0.001$, $\eta_p^2 = 0.98$), and *W* ($F_{2,22} = 344.8$, $p < 0.001$, $\eta_p^2 = 0.97$), but not for $T_{delay}$ ($F_{3,33} = 1.774$, $p = 0.17$, $\eta_p^2 = 0.14$). Posthoc tests showed significant differences for the following *Gap* pairs (at least $p < 0.05$): (0, 72), (0,

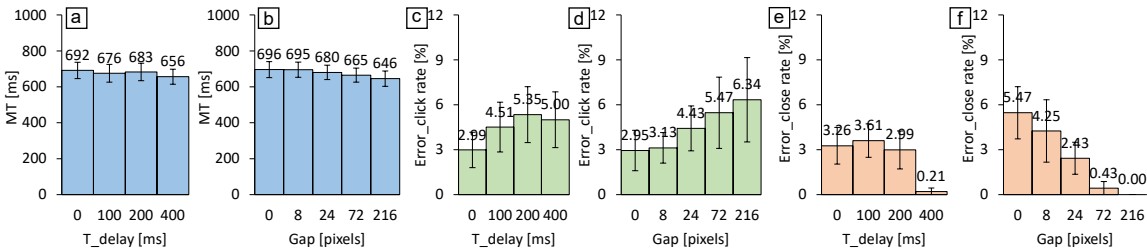

Figure 12: Main effects of $T_{delay}$ and $Gap$ on (a, b) $MT$, (c, d) $ER_{click}$ rate, and (e, f) $ER_{close}$ rate in Experiment 3.

216), (8, 72), (8, 216), (24, 216), and (72, 216). All the $W$ pairs showed significant differences with $p < 0.001$.

Significant interactions were found for $T_{delay} \times A$ ($F_{3,33} = 9.508$, $p < 0.001$, $\eta_p^2 = 0.46$), $Gap \times W$ ($F_{8,88} = 2.397$, $p < 0.05$, $\eta_p^2 = 0.18$), $A \times W$ ($F_{2,22} = 12.23$, $p < 0.001$, $\eta_p^2 = 0.52$), and $Gap \times A \times W$ ($F_{8,88} = 2.138$, $p < 0.05$, $\eta_p^2 = 0.16$). The other combinations of independent variables showed no significant interactions ($p > 0.05$). For the $T_{delay} \times A$ interaction, for both $A$ values, no $T_{delay}$ pairs showed significant differences ($p > 0.05$). For the $Gap \times W$ interaction, significant differences on $Gap$ pairs were found for: (0, 216) and (8, 216) for $W = 25$ pixels, (0, 216), (8, 216), and (24, 216) for $W = 45$ pixels, and (0, 72), (0, 216), (8, 216), and (24, 216) for $W = 75$ pixels. For all $W$ values, significant differences were found for the two $A$ values ($p < 0.001$).

### 5.3.2 $ER_{click}$ Rate

We found significant main effects for $T_{delay}$ ($F_{3,33} = 6.285$, $p < 0.01$, $\eta_p^2 = 0.36$), $Gap$ ($F_{4,44} = 8.088$, $p < 0.001$, $\eta_p^2 = 0.43$), $A$ ($F_{1,11} = 42.36$, $p < 0.001$, $\eta_p^2 = 0.79$), and $W$ ($F_{2,22} = 25.17$, $p < 0.001$, $\eta_p^2 = 0.70$). Posthoc tests showed significant differences for the following $T_{delay}$ pairs (at least $p < 0.05$): (0, 200) and (100, 200). Posthoc tests showed significant differences for the following $Gap$ pairs (at least $p < 0.05$): (0, 24), (0, 72), (0, 216), and (8, 72). $W$ pairs of (25, 45) and (45, 75) showed significant differences with $p < 0.001$.

Significant interactions were found for $T_{delay} \times G$ ($F_{12,132} = 2.970$, $p < 0.01$, $\eta_p^2 = 0.21$), $T_{delay} \times A$ ($F_{3,33} = 5.516$, $p < 0.01$, $\eta_p^2 = 0.33$), $T_{delay} \times W$ ($F_{6,66} = 3.891$, $p < 0.01$, $\eta_p^2 = 0.26$), $Gap \times A$ ($F_{4,44} = 3.595$, $p < 0.05$, $\eta_p^2 = 0.25$), $Gap \times W$ ($F_{8,88} = 2.364$, $p < 0.05$, $\eta_p^2 = 0.18$), $A \times W$ ($F_{2,22} = 6.920$, $p < 0.01$, $\eta_p^2 = 0.39$), $Gap \times A \times W$ ($F_{8,88} = 2.283$, $p < 0.05$, $\eta_p^2 = 0.17$), and $T_{delay} \times Gap \times A \times W$ ($F_{24,264} = 1.826$, $p < 0.05$, $\eta_p^2 = 0.14$). The other combinations of independent variables showed no significant interactions ($p > 0.05$).

For the $T_{delay} \times Gap$ interaction, $T_{delay} = (100, 200)$ showed a significant difference when $Gap = 0$ and $T_{delay} = (0, 200)$ when $Gap = 72$ ($p < 0.05$). For the $T_{delay} \times A$ interaction, $T_{delay} = (100, 200)$ and (200, 400) showed significant differences when $A = 250$ and $T_{delay} = (0, 400)$ when $A = 600$ ($p < 0.05$). For the $T_{delay} \times W$ interaction, $T_{delay} = (0, 200)$, (100, 200), and (200, 400) showed significant differences when $W = 45$ (at least $p < 0.05$). For the $Gap \times A$ interaction, $Gap = (0, 24)$ and (0, 72) showed significant differences when $A = 250$ and $Gap = (0, 216)$ and (8, 216) when $A = 600$ (at least $p < 0.05$). For the $Gap \times W$ interaction, $Gap = (0, 24)$, (0, 72), (0, 216), and (8, 216) showed a significant difference when $W = 45$ and $Gap = (0, 72)$ and (72, 216) when $W = 75$ (at least $p < 0.05$). For $W = 25$ and 75, significant differences were found for the two $A$ values ($p < 0.001$).

### 5.3.3 $ER_{close}$ Rate

We found significant main effects for $T_{delay}$ ($F_{3,33} = 95.59$, $p < 0.001$, $\eta_p^2 = 0.36$), $Gap$ ($F_{4,44} = 75.48$, $p < 0.001$, $\eta_p^2 = 0.87$), $A$ ($F_{1,11} = 171.1$, $p < 0.001$, $\eta_p^2 = 0.94$), and $W$ ($F_{2,22} = 89.95$, $p < 0.001$, $\eta_p^2 = 0.89$). Posthoc tests showed significant differences for the following $T_{delay}$ pairs (at least $p < 0.01$): (0, 200), (0, 400), (100, 200), (100, 400), and (200, 400). Posthoc tests showed significant differences for the following $Gap$ pairs (at least $p < 0.05$): (0, 72), (0, 216), (8, 72), (8, 216), (24, 216), and (72, 216). All $W$ pairs showed significant differences with $p < 0.001$.

Significant interactions were found for $T_{delay} \times Gap$ ($F_{12,132} = 18.54$, $p < 0.001$, $\eta_p^2 = 0.62$), $T_{delay} \times A$ ($F_{3,33} = 73.89$, $p < 0.001$, $\eta_p^2 = 0.87$), $T_{delay} \times W$ ($F_{6,66} = 42.03$, $p < 0.001$, $\eta_p^2 = 0.79$), $Gap \times A$ ($F_{4,44} = 66.65$, $p < 0.001$, $\eta_p^2 = 0.85$), $Gap \times W$ ($F_{8,88} = 33.79$, $p < 0.001$, $\eta_p^2 = 0.75$), $A \times W$ ($F_{2,22} = 124.8$, $p < 0.001$, $\eta_p^2 = 0.92$), $T_{delay} \times Gap \times W$ ($F_{24,264} = 4.100$, $p < 0.001$, $\eta_p^2 = 0.27$), $T_{delay} \times A \times W$ ($F_{6,66} = 3.877$, $p < 0.001$, $\eta_p^2 = 0.26$), $Gap \times A \times W$ ($F_{8,88} = 5.133$, $p < 0.01$, $\eta_p^2 = 0.32$), and $T_{delay} \times Gap \times A \times W$ ($F_{24,264} = 12.36$, $p < 0.001$, $\eta_p^2 = 0.53$). Thus the other combination ($T_{delay} \times Gap \times A$) showed no significant interaction ($p > 0.05$).

For the $T_{delay} \times Gap$ interaction, $T_{delay} = (0, 400)$ and (100, 400) showed significant differences when $Gap = 24$, $T_{delay} = (0, 400)$, (100, 200), (100, 400), and (200, 400) when $Gap = 72$, and $T_{delay} = (0, 400)$, (100, 200), (100, 400), and (200, 400) when $Gap = 216$ (at least $p < 0.05$). For the $T_{delay} \times A$ interaction, $T_{delay} = (0, 200)$, (0, 400), (100, 200), (100, 400), and (200, 400) showed significant differences when $A = 250$ and $T_{delay} = (0, 400)$, (100, 400), and (200, 400) when $A = 600$ ($p < 0.05$). For the $T_{delay} \times W$ interaction, $T_{delay} = (0, 400)$, (100, 400), and (200, 400) showed significant differences when $W = 25$, $T_{delay} = (0, 400)$, (100, 400), and (200, 400) when $W = 45$, and $T_{delay} = (0, 200)$, (0, 400), (100, 200), (100, 400), and (200, 400) when $W = 75$ (at least $p < 0.05$). For the $Gap \times A$ interaction, $Gap = (0, 216)$, (8, 216), (24, 216), and (72, 216) showed significant differences when $A = 250$, and $Gap = (0, 216)$, (8, 24), (8, 72), (8, 216), (24, 216), and (72, 216) when $A = 600$ (at least $p < 0.05$). For the $Gap \times W$ interaction, $Gap = (0, 216)$, (8, 216), (24, 216), and (72, 216) showed significant differences when $W = 25$, $Gap = (0, 216)$, (8, 216), (24, 216), (72, 216) when $W = 45$, and $Gap = (0, 24)$, (0, 72), (0, 216), (8, 72), (8, 216), (24, 216), and (72, 216) when $W = 75$ (at least $p < 0.05$). For all $W$ values, significant differences were found for the two $A$ values ($p < 0.001$).

### 5.3.4 Model Fitting

The $R^2$ values of Fitts' law for each $T_{delay} \times Gap$ condition by using $N = 6$ ($2A \times 3W$) data points ranged from 0.914 to 0.999. When we did not separate the 4 $T_{delay}$ and 5 $Gap$ conditions, the fit using

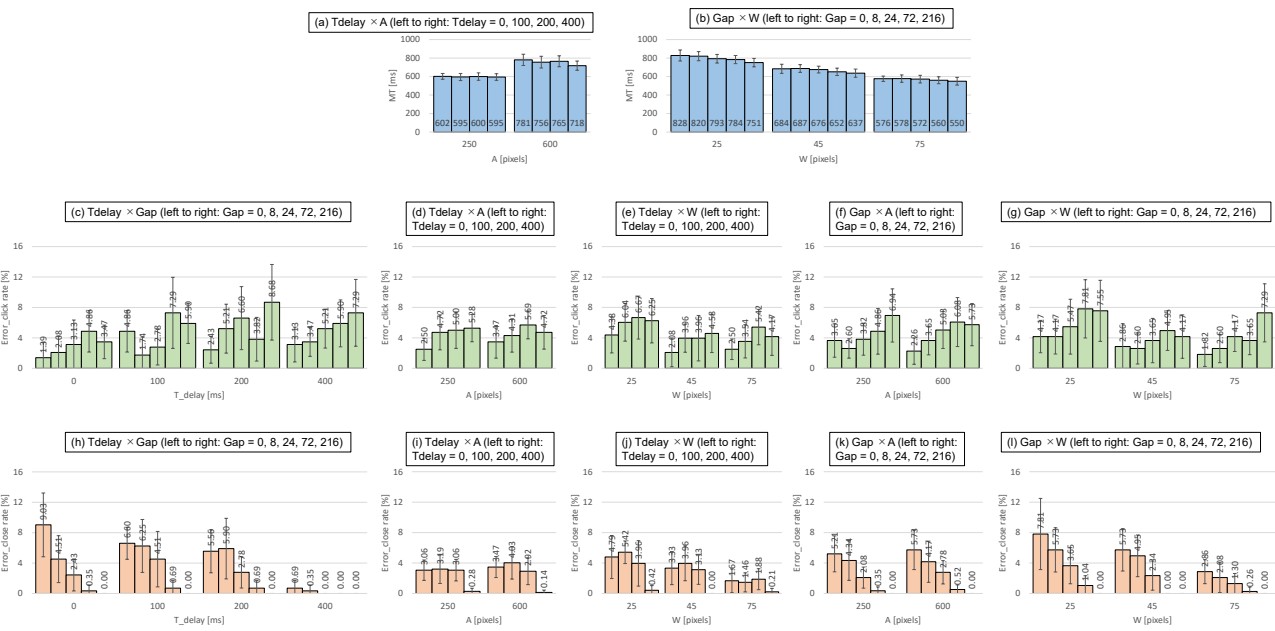

Figure 13: Interactions related to $T_{delay}$ and $Gap$ in Experiment 3.

$N = 120$ data points was $R^2 = 0.930$.

### 5.4 Discussion of Experiment 3

Differently from the result of Experiment 1, $T_{delay}$ showed no main effect on the $MT$. In contrast, a wider $Gap$ significantly decreased the $MT$, which was consistent with Experiment 2. Still, the difference in $MT$ with the minimum and maximum $Gap$ values was small $(696 - 646 = 50$ ms). Also, importantly, our assumption that the $T_{delay}$ and $Gap$ would have a significant interaction for $MT$ was rejected. Therefore, to model the $MT$ data obtained in Experiment 3, it would be sufficient to account for $Gap$ without $T_{delay}$.

In contrast to the $MT$ results, our assumption on the interaction of $T_{delay} \times Gap$ on the $ER_{close}$ rate was confirmed. This result shows that, if the effect to reduce the $ER_{close}$ has already been achieved by either $T_{delay}$ or $Gap$, the other parameter's effect would be limited. For example, for $T_{delay} = 200$ ms in Fig. 13h, the $ER_{close}$ rate for $Gap = 0$ and 8 pixels were still fairly high ($> 5\%$); in comparison, when $T_{delay} = 400$ ms, the $ER_{close}$ rates were almost 0%. This indicates that, for small $Gap$ values such as 0 or 8 pixels, a long $T_{delay}$ is helpful to reduce the $ER_{close}$ rate, but if there is space to set large $Gap$ among GUI items, a long $T_{delay}$ has no benefit.

## 6 GENERAL DISCUSSION

### 6.1 Experimental Design Issue of Learning Effects

For fair comparison, we checked the learning effects on $MT$, $ER_{click}$ rate, and $ER_{close}$ rate in overshoot-prohibited conditions ($T_{delay} = 0$ ms and $Gap = 0$ pixels). The effects of order (Experiments 1 vs. 2) are not significant for $MT$ (a pair-wised two-tailed t-test shows $p = 0.098$) and $ER_{click}$ and $ER_{close}$ rates (Wilcoxon signed-rank tests showed $p = 0.86$ and $p = 0.24$, respectively). For the three participants joining in Experiments 1–3, the effects of order (Experiments 1 vs. 2 vs. 3) on $MT$s and $ER_{close}$ rate are not significant (repeated-measures ANOVA showed $p = 0.48$ and 0.86, respectively). The $ER_{click}$ rate shows $p < 0.05$ but pair-wise tests show no significant differences in all pairs. This result rejects our concerns that learning effects change our conclusions.

### 6.2 Participants' Strategy

While the effects of the $T_{delay}$ and $Gap$ on $MT$, $ER_{click}$ rate, and $ER_{close}$ rate were independently observed in Experiments 1 and 2, the concurrent interaction effects of these factors were not observed for the $MT$ rate in Experiment 3. According to oral interviews, in Experiment 1, 11 of the 12 participants stated that they changed the mouse movement speed depending on the $T_{delay}$. For example, a participant stated as follows.

- "When the timeout [$T_{delay}$] was 100 and 200 ms, I could not immediately return the cursor after entering the pink area in the Exercise Session. Thus, in the actual experiment session, I tried not to overshoot the target as if the timeout was 0 ms. When the timeout was 800 ms and infinity, I did not take care of overshooting the target. When the timeout was 400 ms, I could return the cursor after noticing it entering the pink area in the Exercise, and therefore I was more relaxed to move the cursor than the 800-ms timeout."

Similarly, in Experiment 2, eight participants stated that they changed the speed depending on the $Gap$. A participant stated as follows.

- "I think that the difficulty is categorized into three levels: no gap, medium gap (8, 24 and 72 pixels), and no-effect gap (216 and ∞ pixels). The neighboring condition [$Gap = 0$] was the most difficult. When there was a margin between the target and the distractor, I moved the cursor a little carefully. If the distractor is farthest [$Gap = 216$], I think that the task was the same as no-gap condition [$Gap = \infty$]. So, I intentionally changed my movement speed depending on the gap.

In contrast, in Experiment 3, six and seven participants stated that they changed the speed depending on the $T_{delay}$ and $Gap$, respectively (inclusive). The smaller number of participants who changed the speed may be one reason that there was no significant difference for $T_{delay}$ on $MT$ in Experiment 3.

Hence, not many users purposely adjusted their movement speed depending on the given parameters of $T_{delay}$ and $Gap$ in Experiment 3. Rather, the priority that affected user performance was strongly

on the nominal *ID*. In a real GUI, however, both delays and gaps can be independently designed. Further theoretical development of the model will help GUI designers, but for now, predicting the *MT* by using the baseline Fitts' law model is a suboptimal approach.

In all three experiments, we measured user performance under conditions where the participants knew the $T_{delay}$, and the target and *Gap* areas were explicitly drawn. In *Netflix*, for example, hovering the cursor over a movie thumbnail reveals a *Play* button and detailed descriptions. Similarly to Amazon Prime Video, the circular *Play* button has a *Gap* from the neighboring movie thumbnail *visually*. However, that button has a quite large area that receives mouse-click events, and there is no *Gap* from the neighboring thumbnail in the motor space. In such a case, i.e., where a target has visual and motor spaces of different sizes, user performance would degraded [40, 41], but our findings are limited to a condition of the same visual and motor sizes.

### 6.3 Design Recommendation

The *MT* results of Experiment 1 show that the *MT* was significantly worse for $T_{delay} = 0$ and 100 ms, but the negative effect plateaued for $T_{delay} \geq 200$ ms. In addition, the $T_{delay}$ did not significantly affect the $ER_{click}$ rate, and $T_{delay} \geq 800$ ms did not show a significantly worse $ER_{close}$ rate. Therefore, when there is no gap beyond the target and distractor, using a 800-ms delay to close a pop-up window seems to be needed. If using 400-ms delay, the $ER_{close}$ rate was 2% (Fig. 4c) and there were significant differences from 800 and ∞ ms (both showed 0% $ER_{close}$ rates). It is difficult to recommend using 200-ms delay because it showed the highest $ER_{close}$ rate (∼10%). Considering the tradeoff between low error rate and short waiting time to close pop-up windows, if designers accept the 2% $ER_{close}$ rate, using 400-ms delay seems to be a possible choice. This also justifies the delay configuration for miniature windows shown in Fig. 1b.

Regarding the *Gap*, our results showed that greater gap values were better in terms of decreasing the *MT* and errors in general (Fig. 8 and Fig. 12). On the other hand, a wider gap directly occupies a large space. Hence, our results suggest no clear optimal value for the *Gap* that minimizes wasted spaces while maintaining user performance.

Based on the results of Experiment 3, $T_{delay}$ and *Gap* had a significant interaction on $ER_{close}$ rate, and thus it prevents us from recommending specific values of these parameters. In summary, from the results of Experiments 1–3, we conclude that (1) $T_{delay} = 800$ ms is needed to completely eliminate negative effects of distractor, (2) 400 ms is also possible if designers accept small (2%) unintentional closing of pop-up windows for gap-absent target arrangements, and (3) there is no clear evidence of an optimal *Gap* value. In addition, there remains a possibility that there is more appropriate delay to balance reducing negative effects and shortening a waiting time in untested values around 400 ms, such as 300 or 500 ms.

### 6.4 Limitations and Future Work

Our results cannot be generalized to other conditions besides our experimental setup. For example, cursor responsiveness is affected by mouse-to-cursor transfer functions and by differences between pointing devices such as mouse vs. touchpad [8]. The responsiveness (or *transmission lag*) significantly affects the pointing performance and Fitts' law fitness [18, 29]. A fixed movement direction is also a limitation, as mouse-pointing performance varies depending on directions [48, 50]. Hence, we do not claim that we have found an optimal delay or gap, or that we have found an optimal menu design.

While our tasks were limited to 1D pointing, we are also interested in conducting experiments with a target and distractor of finite heights, which is more realistic for current GUIs (as shown in Fig. 1). From this viewpoint, we feel that our 1D tasks were

somewhat simple compared to real GUIs. Even so, the three experiments provide good motivation for further studies on the topic of overshoot-avoiding pointing performance.

Regarding 1D pointing, we showed page-scrolling tasks as examples in Fig. 2, which are, in actual, dragging-and-dropping operations. Because dragging-and-dropping requires to keep the tension of finger for pressing the mouse button, it degrades user performance in terms of *MT* and error rates [28]. Hence, further studies are needed to confirm if our findings are directly true for dragging and scrolling tasks.

Our future work includes deriving a model in a theoretical way. In our study, the speed profiles did not show clear trends depending on the $T_{delay}$ and *Gap* (Fig. 7 and Fig. 11). Thus, simply adding the $T_{delay}$ and *Gap* terms to Fitts' law would not improve the model fitness. Carefully analyzing the principles involved in changing the movement speed will provide better models to explain both the $T_{delay}$ and *Gap* effects.

In addition, in our experiments, even when an error occurred, a trial was not repeated from the beginning; the participants were asked to immediately aim for the target again. In more realistic scenarios, as described around Fig. 1 and Fig. 2, when we mistakenly click outside the target or overshoot the target, additional time cost to redo the pointing operation is needed. If we adopt such a time-based cost for erroneous operations, the participants would be more careful, and the resultant *MT* would increase [3]. Therefore, we tested only a limited condition where the time-based penalty is 0 ms.

An unclear point regarding GUI designs is that, although 800 ms seems the upper value for ignoring the negative effects of the $T_{delay}$ in Experiment 1, it is possible that such a delay negatively affects users' perceptions and feelings if users want to view the items behind the pop-up window. In addition to the quantitative experiments conducted in this study, subjective evaluation is needed in order to judge whether this value is preferred by users.

## 7 CONCLUSION

We conducted three mouse-pointing experiments in which users had to avoid overshooting a target. User performance was significantly degraded by a shorter delay for acceptance of overshooting and a smaller gap between the target and distractor. Fitts' law held if we run regression expressions for each value of delay and gap, but a more theoretical derivation will be required to capture the effects on *MT*. We found that 800 ms is needed to completely omit negative effects of distractor for densely arranged targets, but we found no optimal gap.

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
