# OpenReview forum: "Evaluating Temporal Delays and Spatial Gaps in Overshoot-avoiding Mouse-pointing Operations"
_graphicsinterface.org/Graphics_Interface/2020/Conference — GI 2020_

### Official Review · AnonReviewer2 · 2020-04-16
**A great amount of effort but the analysis is incomplete and problematic**

**Rating:** 3
**Confidence:** 5

**Review:**

This paper reports on the results of three user studies that explored the effects of temporal delays, spatial gaps, and the combination of delays and gaps on discrete pointing using a mouse. When evaluated independently, temporal delays and spatial gaps were found to increase movement times, however, when evaluated concurrently, an increase in temporal delays resulted in more pop-up window closing errors when gap sizes were small.

Overall, this paper was well written and contains a number of experiments, metrics, statistical results, and Figures. It is quite dense and, although the language is clear, it is difficult to pull out the overarching narrative and contribution to the community. Some of the confusion arises because the full experimental results are not presented. Using Experiment 1 as an example, the text presents the interaction and main effect results for some of the factors but not all (e.g., MT is missing the tDelay x W and tDelay x W x A, ERClick is missing tDelay x W x A, and ERclose is missing tDelay x W x A). For those factors for which interactions were found (for example MT), there are no post-hoc results presented in the text (or details to describe which post-hoc tests were conducted) to explain the tDelay x A and A x W results, yet Figure 4 contains denotations of some post-hoc testing for tDelay and appears to have wrongly collapsed across the A and W factors in the graph (even though there is an A x W interaction effect and there was a tDelay x A but perhaps not a tDelay x W interaction).

Oddly enough, the text itself actually identifies that it is necessary to look at the interaction between factors, i.e., “For example, for Gap = 24 pixels in Figure 11, the ERclose rates are already small (<5%), and thus the ERclose rates did not significantly change even if Tdelay increased. This finding could not have been observed if we had tested only the effect of Tdelay; as shown in Figure 10e, the ERclose rate of Tdelay = 0.4 sec was significantly different from the other three values.” Figure 11 shows the results of the tDelay x Gap interaction for ERclose, which only collapses across the A and W factors (I assume there is no A x W or other interactions, however the text doesn’t describe them), however Figure 10e shows results that have been aggregated over A, W, and Gap – which as Figure 11 shows Gap is important at some tDelay levels so it cannot be aggregated. So, because an incomplete analysis is presented for each metric in each Experiment, an incomplete picture of the actual influence of spatial gaps and temporal delays is presented in this paper and therefore the contribution of this paper is unclear (and likely wrong). (Note that the reference used for the statistical analysis and importance of reporting interaction effects and post-hoc tests comes from Field’s book – one version of which can be found here: https://www.amazon.com/Discovering-Statistics-Using-IBM-SPSS-ebook/dp/B079HYHP5P/ref=sr_1_2?dchild=1&keywords=statistics+ibm+spss&qid=1587070010&sr=8-2 ).

Aside from the issues with the experimental results, I did have a few other questions about the paper.

In the description of the third experiment, the text states that tDelay infinity was removed to evaluate the interaction between tDelay and Gap and then tDelay 0.8 was removed because it was not significantly different from tDelay infinity. In Figure 4, however, there was a significant difference between tDelay 0 and 0.8 on both MT and ER_close and 0.2 and 0.8 on ER_close, so perhaps it should not have been eliminated because it did influence some of the MT and ER_close results. I will also note that the same argument was not applied to the 216 and infinity pixel gaps (which also show a similar pattern of non-significance between 216 and infinity and significance between 8, 24, 72 and 216 for MT and 0, 8, 24 and 216 for ER_close). I wonder what the results of this third experiment would have been, and thus the 400 ms recommendation, if the same exclusion criteria were applied to all the levels of the gap and tDelay factors.

I also was unclear as to the way latency was measured with the mouse – what object was the mouse hit with, how was the velocity of the object controlled over trials, and where did this methodology originate from? The Figure 2 use case also really isn’t ever referenced in the text outside the first few paragraphs of the Introduction so did some text get accidently deleted in the paper or was the inclusion of this Figure in error? I didn’t see any mention of the post-experiment questionnaires that were administered outside of the Participants’ Strategy section – it would be beneficial to include these details in the methodology and is quotes are available, them as well. How long was the third experiment? Lastly, the text refers to both seconds and milliseconds and I found it cumbersome to have to mentally switch between both units – using one unit throughout the text (and one standard abbreviation for milliseconds) would improve readability.

In summary, while I do appreciate the effort and time that has gone in to conducting these three experiments, the analysis that is presented does not provide the level of detail or use an aggregation method necessary to make the types of conclusions that the paper argues is necessary. Because the contribution of the paper comes from the findings, I am thus unable to recommend it for acceptance at this time. I strongly encourage the author(s) to redo the statistical analysis and resubmit their paper because there is value in the experiments they have ran.

---

### Official Review · AnonReviewer3 · 2020-04-20
**Good paper on gaps and delays on mouse-pointing targets, three studies and with Fitt's law evaluation**

**Rating:** 7
**Confidence:** 3

**Review:**

This paper studies the effects of gaps around and time delays on UI elements that are sensitive to overshooting. These elements were studied as the targets of mouse-pointing tasks and evaluated using Fitt's law.

The works is very well motivated by two common usage scenarios: popup elements and scrolling type targeting. Overshooting of hover-based popup elements is a relevant problem as it closes the element and the targeted button becomes unavailable. The same goes for scrolling type interactions, as it is are very tedious to get back to the target once overshot. The authors did a great job in motivating their work, explaining their research question. They clarify that the right gap and delay metrics is not obvious as e.g. larger gaps might reduce overshoot but increase distance and longer delays mitigate overshoot effects but introduce wait times.

The authors conducted three user studies. They evaluate the gap and delay individually, but with the same 12 participants. Their third study investigated interactions between gap and delay and was conducted with mostly new participants, 3 of 12 participants did the previous 2 studies as well. In general, all studies are sound, Bonferroni correction and sphericity tests are reported and the procedure is clearly layed out. However, one question I had is if the order of studies 1 and 2 were counterbalanced to learning effects. The other question was why 8% of the trials were removed for study 2 (page 7)? This should be justified.

In summary, I would recommend accepting this paper as it is very well written, well motivated, relevant, and sound. I also very much appreciate that the authors clearly and openly state that the results cannot be generalized. All design decisions are justified leaving very little open questions.

---

### Official Review · AnonReviewer1 · 2020-04-21
**Deactivation Delay and Around Target Deactivation Gap**

**Rating:** 6
**Confidence:** 5

**Review:**

This paper studies the effect of delayed closing and/or gap to mitigate against overshoot effects in interfaces. The idea is that if targets are sticky (i.e. they don't disappear immediately) or if targets have some tolerance beyond them, then users can increase speed in their movements toward targets, thus improving Fitts's Law performance for targeting tasks.

This is a well-written paper with carefully run experiments. Interestingly, the challenge with this paper is that, in broad strokes, it confirms many of our hypotheses -- that spatial gap around target allows users to be slightly more agressive, that temporal dealy of around 0.4s seems to encourage more agressive pointing behavior that "tops-out" at around this value (see the speed profiles). However, due to between participant speed variances and a small number of participants, the experiments did not show statistically significant effects.

Normally, when I review a paper, I start with big picture ideas, then drill down to some more focused commentary. However, in this case, I actually want to invert my review, talking about some detail through the paper before discussing my current perspective on the paper.

Starting with Related Work, the authors include a section on Latency in Fitts's Law Tasks, but this section is somewhat misleading. Cursor latency is a very different phenomenon from the one explored in this paper. I'd call T_delay in this paper a "deactivation delay", where users have some temporal grace period if they overshoot a transient on-screen target before they are penalized by the target disappearing and then them needing to return to the beginning of the task again, re-activating the transient target, and reaquiring it.  I'm not sure why the latency section was included in this paper -- it's probably worth mentioning, but the only thing I can conclude is that someone in the past might have mentioned this phenomenon, but it is very different, and this section almost seemed a distractor. I would encourage the authors to address this a bit earlier if it was ever a concern, in the intro, and leave it out of related work because it is not directly relevant -- except insofar as cursor latency can result in overshoot.

The studies were largely well-conducted, but I do have some small issues with study design. In particular, the authors state, in Experiment 1, "Even if an erroneous operation was performed, the participants had to immediately aim for the target again; the task was not restarted from the beginning." From my perspective, this actually penalizes their study design, limiting the effects they are looking to identify versus real-world, ecologically valid costs in interfaces.  In real interfaces, if you exceed a delay and/or a gap, then the cost increases because you need to restart the task. Perhaps a friction sound was sufficient for the participants in the study such that their response was appropriately biased simply due to this friction sound, but in the real world the response bias in these tasks is created via temporal cost.

This commentary brings up another point, that of response bias. In psychology, one reading that I might suggest to the authors of this paper is Swetts and Green's work on signal detection in psychological experiments, and in particular the phenomenon of psychological response bias. The idea is that a factor like deactivation delay or gap creates a pre-existing bias in users toward care of agression, and it is this pre-existing user bias that you are trying to explore with your experiment. It isn't necessary to incorporate this into the paper, but it's a useful thing to think -- how psychologists think about these biases.


Another small point in experimental design. When the authors write that "The order of the six T_delay values was balanced among the 12 participants," I assume some sort of Latin Square was used (it's definitely not a full-factorial!).

Finally, in your results and discussion, I'm trying to understand the use of the word suboptimal. It seems to me that, of the values measured, 0.4s was the optimal, trading off performance in various ways -- e.g. in its interaction with gap in experiment 3, for example.  The challenge, here, is that there is not statistical significance beyond 0.1s, and so, given our standards in HCI, it is hard to conclude anything strong from this paper. However, as a demonstration of work-in-progress, I actually think there is something here regarding 0.4s delays. Obviously gap is going to be an advantage if sufficiently large ... It almost seems to me that the "no benefit" beyond 0.1s should be softened to something like "While we see no statistical benefit beyond 0.1s in terms of movement time, it is informative to look at figure 6, where we see that, for delays of 0.4 seconds or larger, the speed profiles differ in peak speed. In fact, one thing I would consider if I were the authors was to include an analysis of peak speed from both studies, to see if there is a difference of peak speed at 0.4s.

However, even if we aren't seeing differences, one problem that the authors may be facing is statistical power due to either the tests they are running or due to the small number of participants. I would encourage the authors to try a linear mixed effects model to see if they can discriminate better with participants as a random effect.

However, even in the absence of statistical effect here, I see value in this paper. I would enourage the authors to do a bit more exploration before it is published (analysis of peak speed, try a more powerful statistical test the RM-ANOVA -- I strongly suggest LME) However, even if these don't result in effects, I would soften some of the language. There is some qualitative evidence here that 0.4s is a good delay (not sub-optimal, perhaps not optimal, but appropriate based on interactions in experiment 3 and speed profiles in experiment 1. At the very least, this work gives a roadmap for more accurately measuring and inferring these effects.

As a result, I'm somewhat positive on this paper. I think it's well written, just a bit too absolute given the somewhat ambiguous results. I am leaning toward accept.

---

### Meta-Review · Area_Chair1 · 2020-04-23

**Recommendation:** Accept
**Confidence:** 5

**Metareview:**

This paper has three detailed reviews, each of which highlights strengths and weaknesses of the paper. Despite the deviation in decision between the three reviewers, there is broad overall agreement that:
- The paper is largely well-written.
- The experiments appear to be conducted appropriately.

As well, all reviewers have suggestions in clarifying language and presentation of results including:
- Some suggestions for additional analyses to clarify whether there truly are no differences between factors (R1).
- Some additional analyses that should be more fully developed in the paper (R3).
- Some additional details on experimental design (R2).
- Some better treatment of latency (R1/3).

Rather than summarize detailed reviews by the external reviewers, I encourage the authors to read the reviews with care and to incorporate the constructive suggestions made by reviewers.

In summary, while I am leaning toward accepting this paper, the authors should carefully consider the points made in each review, perform the additional analyses and report the results with greater care. However, I believe that the authors can modify the paper as per the suggestions of all reviewers such that the paper would rise to the level of acceptability for Graphics Interface.

---

### Decision · Program_Chairs · 2020-04-25

Accept